# Genomic Characterization of *Escherichia coli* O8 Strains Producing Shiga Toxin 2l Subtype

**DOI:** 10.3390/microorganisms10061245

**Published:** 2022-06-17

**Authors:** Xi Yang, Qian Liu, Hui Sun, Yanwen Xiong, Andreas Matussek, Xiangning Bai

**Affiliations:** 1State Key Laboratory of Infectious Disease Prevention and Control, National Institute for Communicable Disease Control and Prevention, Chinese Center for Disease Control and Prevention, Beijing 102206, China; yangxicdc163@163.com (X.Y.); qianliu9608@gmail.com (Q.L.); sunhui@icdc.cn (H.S.); xiongyanwen@icdc.cn (Y.X.); 2Division of Clinical Microbiology, Department of Laboratory Medicine, Karolinska Institute, 141 52 Stockholm, Sweden; anmatu@ous-hf.no; 3Division of Laboratory Medicine, Oslo University Hospital, 0372 Oslo, Norway; 4Division of Laboratory, Medicine Institute of Clinical Medicine, University of Oslo, 0372 Oslo, Norway

**Keywords:** Shiga toxin, *Escherichia coli*, Stx2l, whole genome sequencing

## Abstract

Shiga toxin-producing *Escherichia coli* (STEC) can cause diseases ranging from mild diarrhea to fatal extra-intestinal hemolytic uremic syndrome (HUS). Shiga toxin (Stx) is the key virulence factor in STEC, two Stx types (Stx1 and Stx2) and several subtypes varying in sequences, toxicity, and host specificity have been identified. Stx2l is a newly-designated subtype related to human disease but lacks thorough characterization. Here, we identified Stx2l from five STEC strains (Stx2l-STECs) recovered from raw mutton and beef in China. Whole-genome sequencing (WGS) was used to characterize the Stx2l-STECs in this study together with Stx2l-STECs retrieved from public databases. Our study revealed that all the analyzed Stx2l-STEC strains belonged to the same serogroup O8. Multilocus sequencing typing (MLST) showed two sequence types (ST88 and ST23) among these strains. Stx2l-converting prophages from different sources shared a highly similar structure and sequence. Single-nucleotide polymorphism (SNP)-based analysis revealed genetic relatedness between the human-derived and food-derived strains belonging to ST23. To conclude, our study supported the designation of Stx2l and demonstrated diverse host range and geographical distribution of Stx2l-STECs.Stx2l-STEC strains from different sources showed a high genetic similarity with an identical O8 serogroup. Further studies are needed to investigate the epidemiological trait and pathogenic potential of Stx2l-STEC strains.

## 1. Introduction

Shiga toxin-producing *Escherichia coli* (STEC) is a significant foodborne pathogen that can cause human gastrointestinal diseases ranging from mild diarrhea to life-threatening hemolytic uremic syndrome (HUS) [1,2]. The key virulence factor of STEC is Shiga toxin (Stx), which is encoded by *stx* located downstream of lysogenized lambdoid prophages [3,4]. Stx-converting prophages are highly mobile genetic elements that play an important role in horizontal gene transfer and STEC pathogenesis [5]. Stx comprises two immunologically distinct types (Stx1 and Stx2) [6]. Based on variations in amino acid sequences, a standardized taxonomic nomenclature proposed by Scheutz et al. distinguished Stx1 and Stx2 into various subtypes, i.e., three Stx1 (Stx1a, Stx1c, and Stx1d) and seven Stx2 (Stx2a to Stx2g) subtypes [7]. Different Stx subtypes vary in biologic activity leading to a difference in epidemiological association with patient outcomes. Stx2-producing strains are more often associated with severe disease than Stx1-producing strains [8,9]. Since the establishment of the standardized Stx subtyping approach, several novel Stx2 subtypes have been reported, including Stx2h to Stx2m and Stx2o [10,11,12,13,14]. The newly proposed Stx2l subtype, initially designed as Stx2e, has been identified in a few clinical and sheep isolates [9,15]. However, the characteristics of the Stx2l-STEC strains have been poorly elucidated.

In the present study, we identified the Stx2l subtype from raw meats-derived STEC strains in China and strains of diverse origins from other countries. Whole-genome sequencing (WGS) was performed to characterize the genomic features of Stx2l-STEC strains and Stx2l-converting prophages. The phylogenetic relatedness of the Stx2l-STEC strains in this study and those reported from other sources was assessed.

## 2. Materials and Methods

### 2.1. Ethics Statement

The current study was reviewed and approved by the ethics committee of the National Institute for Communicable Diseases Control and Prevention, China CDC, with the number ICDC-2017003.

### 2.2. Identification of Stx2l Subtype from STEC Strains

Identification of STEC strains carrying *stx2l* subtype (Stx2l-STEC) from our STEC collection was performed by an in-house *stx*_subtyping approach based on WGS data as previously described [16,17]. Briefly, an in-house *stx*_subtyping database was created including representative nucleotide sequences of all identified Stx1 and Stx2 subtypes. Notably, the originally-designated Stx2e variant Stx2e-O8-FHI-1106-1092 (AM904726.1) was redesignated as Stx2l [9]. Stx2l-STEC were identified by screening genome assemblies against the updated *stx*_subtyping database using ABRicate version 0.8.10 (https://github.com/tseemann/abricate) (accessed on 1 April 2022). Other Stx2l-STEC genomes were retrieved from the National Center for Biotechnology Information (NCBI, https://www.ncbi.nlm.nih.gov/) and EnteroBase databases (https://enterobase.warwick.ac.uk/species/ecoli) (all accessed on 1 April 2022).

To validate the taxonomic position of Stx2l, the full nucleotide sequences of Stx2l in this study were extracted from the genome assemblies; several representative nucleotide sequences of all the Stx2 subtypes (Stx2a to Stx2m and Stx2o) and variants were selected as references and downloaded from the GenBank. The amino acid sequences for the combined A and B subunits of Stx2 holotoxin were translated from the open reading frames. Phylogenetic trees based on the holotoxin amino acid sequences were reconstructed with three algorithms, Neighbor-Joining, Maximum Likelihood, and Maximum Parsimony, using MEGA 11 software (www.megasoftware.net) (accessed on 5 April 2022); the stability of the groupings was estimated by bootstrap analysis (1000 replications).

### 2.3. WGS-Based Molecular Characterization of Stx2l-STEC Strains

In silico serotyping was conducted by comparing genome assemblies against the SerotypeFinder database (https://cge.food.dtu.dk/services/SerotypeFinder/) (accessed on 10 April 2022) using ABRicate version 0.8.10 with default parameters. Multilocus sequence typing (MLST) was performed in silico through the online tool provided by the Warwick *E. coli* MLST scheme website (https://enterobase.warwick.ac.uk/species/ecoli/allele_st_search) (accessed on 10 April 2022).

### 2.4. Genomic Characterization of Stx2l-Converting Prophages

Complete genomes of the Stx2l-STEC strains were uploaded to the PHAge Search Tool Enhanced Release (PHASTER, http://phaster.ca/) (accessed on 12 April 2022) to identify Stx-converting phages. The intact Stx2l prophage sequences were extracted from the complete STEC genomes. To obtain the Stx2l prophages from the draft genomes, the RAST server [18] was used to annotate the draft Stx2l-STEC genomes; the sequences of the Stx2l-converting prophages from those draft genomes were reconstructed from multiple contigs based on BLASTn searching, RAST annotation, and progressive Mauve alignment [19]. Intact Stx2l prophages were defined when complete prophage structures were identified. The gene adjacent to the integrase gene was designated as the phage insertion site [4]. Stx2l-converting prophages were compared and visualized using Easyfig [20].

### 2.5. Single Nucleotide Polymorphism (SNP)-Based Phylogeny of Stx2l-STECs

A whole-genome SNP phylogeny was used to assess the genomic relationship of the Stx2l-STEC strains reported so far. The core alignment of the SNPs was obtained by using snippy-multi in Snippy version 4.3.6 (https://github.com/tseemann/snippy) (accessed on 15 April 2022) with the default parameters; the Stx2l-STEC strain STEC306 in this study (SAMN21841557) was used as a reference. Gubbins version 2.3.4 [21] was then used to remove recombination from core SNP alignments and construct a maximum likelihood tree based on the filtered SNP alignments. The SNP distances were executed using snp-dists v0.7.0. (https://github.com/tseemann/snp-dists) (accessed on 15 April 2022). The visualization and annotation of the phylogenetic tree were produced by the web-tool ChiPlot (https://www.chiplot.online/#Phylogenetic-Tree (accessed on 18 April 2022).

### 2.6. Data Availability

The genome data presented in this study are publicly available in NCBI with the accession numbers SAMN20824184, SAMN21841566, SAMN21841557, SAMN21841558, SAMN21841559, SAMN16454166, SAMN12290412, SAMEA2593965 and SAMN05171628.

## 3. Results

### 3.1. Identification of Stx2l Subtype in Food-Derived STEC Strains

The in-house *stx* subtyping approach showed that the *stx2* sequences of five isolates in our STEC strains collection (n = 882) shared 99.68% nucleic acid sequence identity with the Stx2l subtype. Three strains were isolated from raw mutton samples collected in a market in Beijing, China, in December 2013. Two strains were isolated from one raw beef and one raw mutton sample collected in the same market in March 2014. These isolates were previously identified as Stx2e-STEC strains based on the nomenclature proposed in 2012 [7]. We performed BLASTn searching using the *stx2* reference sequence (AM904726.1) against the Refseq Genome Database (taxid:562); *stx2* sequences from six STEC genomes yielded a nucleic acid sequence identity above 99% with the reference *stx2l* sequence. In addition to the five raw meat-derived strains in this study, two food-derived STEC strains from the USA, and one gastroenteritis patient-derived STEC strain from Norway were found to carry *stx2l*. One human-derived Stx2l-STEC strain from the UK was found in the EnteroBase database. Thus, nine Stx2l-STEC genomes were included in the subsequent analysis.

Phylogenetic trees based on the holotoxin amino acid sequences of all Stx2 subtypes using the Neighbor-Joining (Figure 1), Maximum-Likelihood, and Maximum Parsimony algorithms shared the same topology (data not shown); the Stx2 of the five strains in this study and four strains from other sources clustered with Stx2l type sequence Stx2l-O8-FHI-1106-1092, while they formed a distinct lineage from other Stx2 subtypes (Figure 1). These data supported that these STEC strains harbor the *stx2l* subtype. Seven out of the nine strains shared identical Stx2l amino acid sequence, while one and two amino acid differences were found in the Stx2l representative strain FHI-1106-1092 and strain 879916 from food, respectively, when compared with the others. Identical IS2 family transposase was inserted into the intergenic region between the A and B subunits of Stx2 from three Stx2l-STEC isolates (STEC306, STEC307, and STEC308); the intergenic regions among other six *stx2l* sequences were identical, which contained 11 nucleotides (aggagttaagt).

### 3.2. Stx2l-STEC Strains Belonged to Serogroup O8

The WGS data analysis revealed an identical O8 serogroup of the nine Stx2l-STEC strains. Two meat-derived Stx2l-STEC strains in this study, two food-derived strains from the USA, and one gastroenteritis patient strain from Norway were assigned to the same serotype O8:H30, while three meat-derived Stx2l-STEC strains in this study and one human-derived strain from the UK were assigned as O8:H9. Of note, all food-derived strains belonged to the same MLST type ST88, while two human strains belonged to the MLST type ST23.

### 3.3. Genetic Feature of Stx2l-Converting Prophages

The BLASTn search showed that the Stx2l-STEC strain STEC306 (SAMN21841557) harbored the most similar phage sequence with the other Stx2l prophages; thus, this complete Stx2l-converting prophage was selected as a reference to identify prophages from the draft genomes. Seven complete and two incomplete Stx2l prophages were obtained. The comparison of the Stx2l prophages demonstrated the genetic similarity among these prophages from different sources (Figure 2). Similar to other Stx prophages [22], three major modules, i.e., the integration cassette, lysis cassette, and morphogenetic related genes, were found in the Stx2l-converting prophages; the sequence and structure of the three modules each were nearly identical among the different Stx2l-converting prophages. Variability was observed at their insertion sites and phage integrase; Stx2l prophages of the four food-derived strains were inserted at *ypjA* (adhesin-like autotransporter YpjA/EhaD), while the human-derived Stx2l prophages and three food-derived Stx2l prophages were inserted at *parB* (ParB/RepB/Spo0J family partition protein). In addition to the identical integrase (WP_033813161.1) possessed by all Stx2l prophages, the recombinase family protein (WP_000135615.1) and integrase domain-containing protein (WP_248232026.1) were found in the human-derived Stx2l prophages and other three food-derived Stx2l prophages.

### 3.4. SNP-Based Phylogenetic Relationship of Stx2l-Producing Strains

To assess the phylogenetic relationships of the Stx2l-STEC strains, whole-genome SNP-based phylogeny trees were constructed. The SNP analysis identified 1491 SNPs among the nine Stx2l-STEC strains. Two distinct clusters were observed based on the MLST types (ST88 and ST23) of strains (Figure 3). Food-derived isolates from China and USA formed a cluster, raw mutton- and human-derived Stx2l-STEC strains from different countries were clustered. Pairwise SNP distance heatmaps were produced to illustrate the dissimilarity among strains. The SNP distance between the three food- and two human-derived strains in the ST23 cluster was ≤ 179, and it was ≤ 266 among four ST88 food-derived strains from China and USA.

## 4. Discussion

To our knowledge, this is the first study reporting the molecular traits of Stx2l-STEC strains. Different Stx subtypes vary in host specificity and toxicity, resulting in variations in pathogenic potentials to humans. Thus, precise Stx subtyping is valuable for risk assessment at an early stage after STEC infection. Since the standardization of Stx nomenclature and emergence of new Stx variants, novel Stx subtypes have been identified and a few Stx subtypes have been redesignated. The provisionally designed Stx2e variant Stx2e-O8-FHI-1106–1092 was redefined as Stx2l [7]. Therefore, we screened our STEC collection using the updated *stx*_subtyping database and identified five strains carrying the *stx2l* subtype, which was previously identified as *stx2e* based on the earlier nomenclature [7]. The sequence of transposable element IS2 was found among three of the five *stx2l* [17]. By searching the literature and publicly available sequences, several strains were found to carry the *stx2l* subtype, including the five meat-derived strains in this study, diarrheal patients-derived strains in Norway and Denmark [9], a human-derived strain in the UK, Roquefort cheese-derived strains in the USA [11], and sheep-derived strains in Ireland [15]. These data demonstrated a wide host range and geographical distribution of Stx2l-STEC strains.

We characterized the molecular characteristics of the Stx2l-STEC strains using WGS. The identical serogroup O8 was found among the five raw meat-derived strains in this study and all the publicly available genomes of Stx2l-STEC strains; notably, this serogroup was also possessed by the Stx2l reference strain FHI-1106-1092 and the patients-derived strains in Norway and Denmark [9], suggesting that O8 might be the dominant serogroup in the Stx2l-STEC strains. It should be noted that the O8:H30 Stx2l-STECs in this study showed a close phylogenetic relationship with the food-derived strains from USA; strains of this serotype have been isolated from diarrheal patients [9], suggesting the pathogenic potential of O8:H30 Stx2l-STEC strains. The whole-genome phylogeny showed that the patients-derived Stx2l-STEC strains from Norway and UK clustered with the three food-derived strains in this study; all these strains belonged to the MLST type ST23, indicating Stx2l-STECs of ST23 might be spread and, thus, possibly pose global public health risk. However, more Stx2l-STECs strains are needed to gain further insights. We further characterized the seven complete and two incomplete Stx2l-converting prophages; our data demonstrated a high similarity of the Stx2l prophages from different sources. It should be mentioned that the three mutton-sourced Stx2l-STECs shared nearly identical genomes; as these strains were isolated from different samples collected in the same sampling site, it is likely that they derived from the same clone.

In conclusion, our study demonstrated a potential wide distribution of Stx2l-STEC in diverse hosts and geographical regions. The genomic characterization revealed the genetic similarity of the Stx2l-STEC strains from different sources, with O8 possibly being the predominant serogroup. The genomes of the Stx2l-converting prophages from different sources were conserved. The Stx2l-STEC strains were phylogenetically clustered based on the sequence type of strains, and strains with ST23 might pose a public health risk.

## Figures and Tables

**Figure 1 microorganisms-10-01245-f001:**
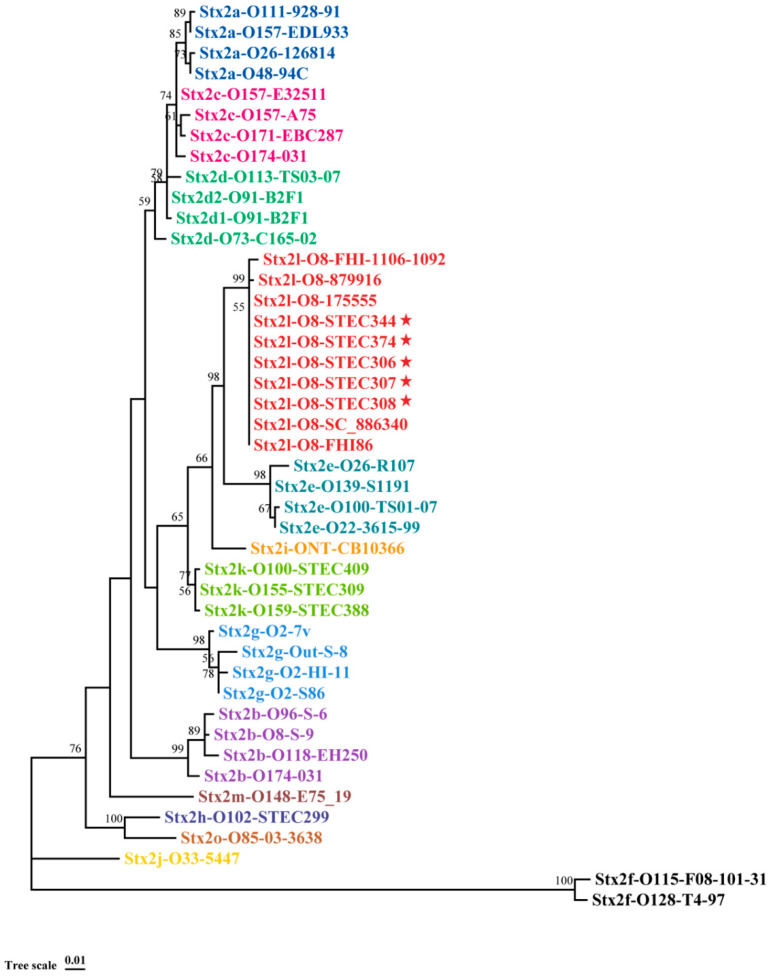
Phylogenetic tree of the Stx2 subtypes by the Neighbor-Joining method. The Neighbor-Joining tree was inferred from comparison of the combined A and B holotoxin amino acid sequences of all Stx2 subtypes. The numbers on the tree indicate the bootstrap values calculated for 1000 subsets for branch points > 50%. Tree scale, 0.01 substitutions per site. Stx2 subtypes are indicated by different colors. The red stars indicate strains in this study.

**Figure 2 microorganisms-10-01245-f002:**
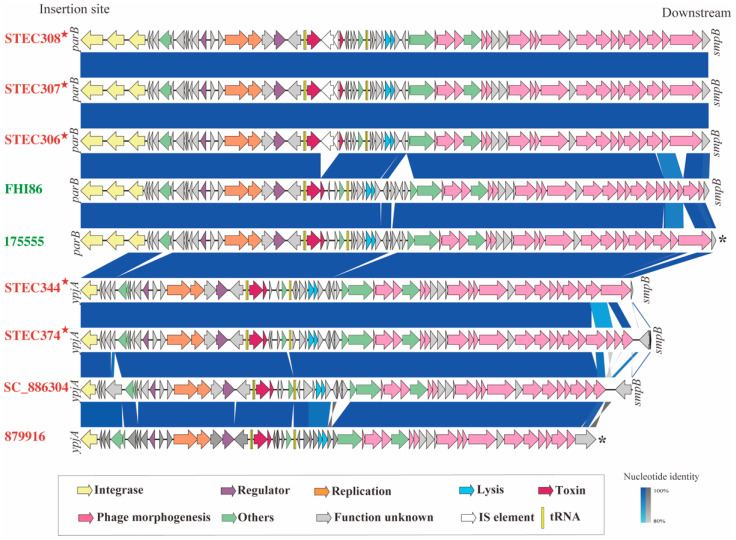
Comparison of the nine Stx2l-converting prophages. The arrows indicate gene directions. The coding sequences are represented by arrows and linked by blue bars shaded to represent the nucleotide identity and gray bars shaded to represent the nucleotide identity of reverse complemented homologous sequences, as indicated in the legend. The color of the text indicates the source of the strains; red represents food-derived Stx2l-STEC strains, and green represents human-derived Stx2l-STEC strains. The red stars indicate strains in this study. The insertion sites and downstream of prophages in the chromosomes are shown at the start and end of the prophages. The asterisk (*) signifies that downstream of the phage could not be conclusively identified.

**Figure 3 microorganisms-10-01245-f003:**
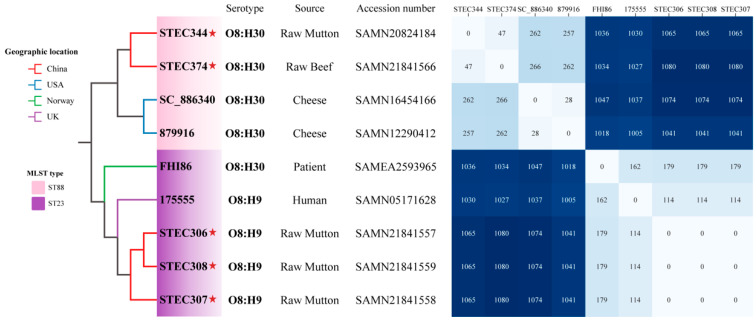
Phylogenetic tree based on single-nucleotide polymorphisms (SNPs) using the Maximum-Likelihood method. Branch length is ignored for better visualization. The red stars indicate strains in this study. The geographic location, MLST type, serotype, source, and accession number of all strains are shown. The heatmap illustrates pairwise SNP distances between the Stx2l-STEC strains.

## Data Availability

The genome data presented in this study are publicly available in NCBI with the accession numbers SAMN20824184, SAMN21841566, SAMN21841557, SAMN21841558, SAMN21841559, SAMN16454166, SAMN12290412, SAMEA2593965 and SAMN05171628.

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
