# Peer review of "Genomic Characterization of Escherichia coli O8 Strains Producing Shiga Toxin 2l Subtype"

_microorganisms, 2022, doi:10.3390/microorganisms10061245_

Round 1
Reviewer 1 Report
Comments microorganisms-1768962
The manuscript entitled “Genomic Characterization of Escherichia coli O8 Strains Producing Shiga Toxin 2l Subtype” is a short report on the genome comparison of nine E. coli STEC strains carrying the Stx21 toxin variant, of which five strains were from the author's institution. The strains were assigned to Sts and to O:H antigen types. Since strains in one SNP phylogenetic branch included patiet isolates, the authors conclude that Stx21 E. coli can have clinical significance. The study is well written and has a clear scope, so that I endorse publication after minor revision according to the comments below:
Some additional points should be commented in the discussion: the SNP distances observed and completeness of toxin converting prophages;
Lines 209-210; unclear statement. Please, make it more explanatory
Minor points:
Check that spaces are correctly placed (e.g. Line 53)
Lines 234-235; need to be analyzed
Reviewer 2 Report
Authors present data on the clinically relevant Shiga toxin-producing Escherichia coli strains, which have been recovered from raw mutton and beef, and produces a novel Shiga toxin subtype – Stx2l. In addition, the Stx2l-converting prophages from the tested strains are bioinformatically characterized. The obtained results are important for understanding of evolution of pathogenic variants of Escherichia coli as well as for upgrading of epidemiological surveillance of STEC infections and to customize a clinical diagnosis.
The presented work shows every sign of the original study and demonstrates the required novelty. The experimental setup and description of methods seems adequate for the field. All tools and methods employed are appropriately documented. The results are presented clearly, with the appropriate controls and statistical analysis. The appropriate illustrations demonstrating the main achievements are offered.
Author Response
We sincerely thank the reviewer’s positive evaluation on our manuscript.